# Optimisation of the Microwave-Assisted Ethanol Extraction of Saponins from Gac (*Momordica cochinchinensis* Spreng.) Seeds

**DOI:** 10.3390/medicines5030070

**Published:** 2018-07-03

**Authors:** Anh V. Le, Sophie E. Parks, Minh H. Nguyen, Paul D. Roach

**Affiliations:** 1School of Environmental and Life Sciences, University of Newcastle, Ourimbah, NSW 2258, Australia; sophie.parks@dpi.nsw.gov.au (S.E.P.); Minh.Nguyen@newcastle.edu.au (M.H.N.); Paul.Roach@newcastle.edu.au (P.D.R.); 2Faculty of Bio-Food Technology and Environment, University of Technology (HUTECH), HCMC 700000, Vietnam; 3Central Coast Primary Industries Centre, NSW Department of Primary Industries, Ourimbah, NSW 2258, Australia; 4School of Science and Health, Western Sydney University, Penrith, NSW 2751, Australia

**Keywords:** Gac seeds, *Momordica cochinchinensis*, saponins, microwave-assisted extraction, optimization

## Abstract

**Background**: Gac (*Momordica cochinchinensis* Spreng.) seeds contain saponins that are reportedly medicinal. It was hypothesised that the extraction of saponins from powdered Gac seed kernels could be optimised using microwave-assisted extraction (MAE) with ethanol as the extraction solvent. The aim was to determine an appropriate ethanol concentration, ratio of solvent to seed powder and microwave power and time for extraction. Whether or not defatting the Gac seed powder had an impact on the extraction of saponins, was also determined. **Methods**: Ethanol concentrations ranged from 60–100% were used to compare total saponins content (TSC) extracted from full-fat and defatted Gac seeds. Ratios of solvent to Gac seeds ranged from 10 to 100 mL g^−1^ and microwave conditions ranged from 1–4 cycles at power levels ranged from 360–720 W, were examined successively to evaluate their efficiency in extracting saponins from full-fat Gac seeds. **Results**: A four-fold higher of TSC was obtained in extracts from full-fat Gac seed powder than from defatted powder (100 vs. 26 mg aescin equivalents (AE) per gram of Gac seeds). The optimal parameters for the extraction of saponins were a ratio of 30 mL of 100% absolute ethanol per g of full-fat Gac seed powder with the microwave set at 360 W for three irradiation cycles of 10 s power ON and 15 s power OFF per cycle. **Conclusions**: Gac seed saponins could be efficiently extracted using MAE. Full-fat powder of the seed kernels is recommended to be used for a better yield of saponins. The optimised MAE conditions are recommended for the extraction of enriched saponins from Gac seeds for potential application in the nutraceutical and pharmaceutical industries.

## 1. Introduction

*Momordica cochinchinensis* Spreng. is a perennial climber, which belongs to the Cucurbitaceae family. It ranges from China to the Moluccas and has been used in food and traditional medicine in East and Southeast Asia [1]. The most important part of the mature fruit is the red flesh surrounding the seeds, called the aril, which is used as a colorant in rice or as a material for further processing into functional food ingredients. The seeds are not eaten and they are removed from the aril and are mostly considered waste [2]. However, in traditional medicine, Gac seeds are alleged to have a wide array of therapeutic effects for a wide variety of conditions, including fluxes, liver and spleen disorders, hemorrhoids, wounds, bruises, inflammation, swellling and infections [1,3]. Modern science has reported biological activities for Gac seed extracts, including being a gastroprotective agent [4,5] and accelerating the healing of gastric ulcers in rats [6], and possessing antitumour [7], anticancer [8] and anti-inflammatory [9,10] activities.

Gac seed saponins have been reported to be critical constituents in Gac seed extracts, which were responsible for their medicinal properties [9,11]. These constituents of Gac seeds have been investigated by several investigators: two saponins, referred to as momordica saponin I and II, have been isolated and characterised [12], in which momordica saponin I is a major gastroprotective ingredient [5]. Another saponin, karounidiol, a compound possessing cytotoxic activity against human cancer cell lines [13], has been reported to be present in Gac seeds [14]. The potential valuable pharmaceutical properties of the Gac seed saponins warrants investigating how they are best extracted from the seeds i.e., which extraction technique(s) will maximise the yield of saponins.

The conventional extraction technique, in which the solid material is suspended in extraction solvent with no assistance for breaking the cell structure of the solid material, is often associated with a long heating time, which risks the degradation of bioactive compounds. This has led to the proposed use of advanced techniques such as microwave-assisted extraction (MAE) and ultrasonic-assisted extraction (UAE) that are efficient in terms of extraction time and solvent consumption. Microwave heating or ultrasonic cavitation is able to disrupt the plant cell structure via an increase in the internal pressure of the cell and thereby, release the bioactive compounds [15,16]. However, in a comparative study being carried out by the same authors [17], it was found that while the MAE significantly improved Gac seed saponin extraction in comparison to the conventional method, UAE did not. MAE, therefore, is the technique which needs to be further optimised. The MAE method is likely to be effective for the extraction of saponins from the Gac seeds, as it has been reported that microwave assistance significantly improved the recovery of saponins from a wide range of plant sources such as *Phyllanthus amarus* [18], yellow horn [19], *Ganoderma atrum* [20], chick pea [21] and ginseng [22], among others.

The choice of the extraction solvent is also important. Low alcohols such as methanol and ethanol have usually been used as effective solvents for the extraction of saponins from plant materials. However, according to the US Food and Drug Administration [23], methanol belongs to the Class 2 solvents, which should be limited in pharmaceutical products because of their inherent toxicity. Ethanol, on the other hand, belongs to the Class 3 solvents [23], which are less toxic and of lower risk to human health and therefore, should be used instead of methanol for the extraction of plant bioactive compounds. Moreover, ethanol in form of wines has been traditionally used for maceration of Gac seeds, therefore, it is reasonable to investigate the efficiency of this solvent for modern extraction methods. In addition, ethanol is also an excellent microwave absorbing solvent and has been used to advantage in MAE [16].

When it comes to extraction of saponins from seeds, defatting is often carried out before the saponins are extracted [24]. Although the defatting might make it simpler for the saponin extraction in terms of technique, and does not greatly affect the saponin yield for some type of seeds, it can cause a great loss of saponin for others.

Therefore, in this study, the extraction of saponins from powdered Gac seed kernels was optimised using MAE with ethanol as the extraction solvent. The aim was to determine an appropriate ethanol concentration, ratio of solvent to seed powder and microwave power and time for saponin extraction. Whether or not defatting the Gac seed powder had an impact on the extraction of saponins, was also determined.

## 2. Materials and Methods

### 2.1. Materials

#### 2.1.1. Solvents, Reagents and Chemicals

Absolute ethanol (≥99.8%), methanol and chemicals including vanillin, sulphuric acid, and potassium persulfate were products of Merck (Bayswater, VIC, Australia) and 2,4,6-tris(2-pyridyl)-*s*-triazine; (±)-6-hydroxy-2,5,7,8-tetramethylchromane-2carboxylic acid (trolox), aescin, 2,2-diphenyl-1-picrylhydrazyl (DPPH), 2,2′-Azino-bis(3ethylbenzothiazoline-6-sulfonic acid) diammonium salt (ABTS) were products of Sigma-Aldrich Co. (Castle Hill, NSW, Australia).

#### 2.1.2. Gac Seed Kernel Powder

Gac seeds, were collected from 450 kg of fresh Gac fruit, from accession VS7 as classified by Wimalasiri, Piva, Urban and Huynh [25]. These fruits were bought at Gac fruit farms in Dong Nai province, Ho Chi Minh city, Vietnam (Latitude: 10.757410; Longitude: 106.673439). After their separation from the fresh fruit, the seeds were vacuum dried at 40 °C for 24 h to reduce moisture and increase the crispness of the shell to facilitate shell removal. The dried seeds were de-coated to obtain the kernels, which were then packaged in vacuum-sealed aluminum bags and stored at −18 °C prior to use.

#### 2.1.3. Preparation of Gac Seed Kernel Powder

The Gac seed kernels were ground in an electric grinder (100 g ST-02A Mulry Disintegrator), to produce powder, which could pass through a sieve of 1.4 mm. The powder was then freeze-dried using a Dynavac FD3 freeze dryer (Sydney, NSW, Australia) for 48 h at −45 °C under vacuum at a pressure loading of 10^−2^ mbar (1 Pa) to reduce the moisture content to 1.21 ± 0.02%, as determined using a MOC63u moisture analyser (Shimazdu, Kyoto, Japan). This Gac seed kernel powder was referred to as ‘full-fat powder’ and was stored in vacuum-sealed polyethylene bags under vacuum at −20 °C until used.

#### 2.1.4. Preparation of Defatted Gac Seed Kernel Powder

To prepare defatted Gac seed kernel powder, the freeze-dried kernel powder was extracted three times for thirty minutes with hexane (1:5 *w*/*v*) on a magnetic stirrer at room temperature. Each time, the resulting slurry was suction-filtered and the final residue was air-dried for 12 h and stored in a desiccator at ambient temperature until used. This Gac seed kernel powder was referred to as ‘defatted powder’.

### 2.2. Methods

The experimental design for the study is shown in Figure 1.

#### 2.2.1. Microwave Assisted Extraction (MAE)

The MAE was performed using a R395YS Sharp Carousel microwave oven (Sharp Corporation, Bangkok, Thailand) bought from a local Target store (Tuggerah, NSW, Australia). Gac seed kernel powder was mixed with ethanol of various concentrations with water in a 100 mL conical flask. The suspension was left pre-leaching for 30 min at ambient temperature before microwave treatment was applied for varying number of cycles, which consisted of 10 s power ON and 15 s power OFF per cycle. The temperature of the suspension was recorded at the end of the MAE process.

#### 2.2.2. Extraction of Saponins from Full-Fat and Defatted Gac Seed Kernel Powders

Prior to weighing for extraction, the moisture content of the powder samples was measured using a MOC63u moisture analyser (Shimazdu, Kyoto, Japan), which was used in the determination of saponin yield.

The effect of the ethanol concentration was investigated for the MAE of saponins from both the full-fat and the defatted powders. The concentration of ethanol was varied (60%, 70%, 80%, 90% and 100%) but the solvent to powder ratio and the microwaving conditions were kept constant at 30 mL g^−1^ and 600 W for four cycles, respectively (1st experiment in Figure 1). After finishing the extractions, the suspensions were rapidly cooled to ≤20 °C in an ice water bath and filtered through a 0.45 µm membrane filter. The clear extracts were collected and kept at −20 °C for analysis within a week.

#### 2.2.3. Extraction of Saponins from the Full-Fat Seed Kernel Powder

The full-fat powder was selected for the following two experiments since it resulted in a higher extraction of saponins for all the concentrations of ethanol; and 100% absolute ethanol was chosen because it resulted in the highest extraction of saponins from the full-fat powder.

Two experiments (Figure 1) were done using the full-fat powder and 100% ethanol as the extraction solvent to determine the effect of three individual parameters, (i) the ratio of solvent to powder (10, 20, 30, 40, 60, 80 and 100 mL g^−1^) (Figure 1, 2nd Experiment), (ii) microwave radiation power (360, 480, 600, 720 and 840 W) and (iii) microwave irradiation time (1, 2, 3 and four cycles) (Figure 1, 3rd Experiment), on the recovery of saponins from the Gac seed kernel powder was investigated. When one parameter was examined, the other was maintained constant; for the 2nd experiment (Figure 1), the microwave conditions were 600 W with four cycles and for the 3rd experiment, the ratio of ethanol to powder was 30 mL g^−1^. After finishing the extractions, the suspensions were rapidly cooled to ≤20 °C in an ice water bath and filtered through a 0.45 µm membrane filter. The clear extracts were collected and kept at −20 °C for less than a week before analysis.

#### 2.2.4. Verifying Optimal Conditions for Gac Seed Saponin Extraction

From the findings in the 3rd experiment, two possible optimal sets of microwave parameters were chosen for the extraction of saponins from the full-fat powder. Therefore, these two sets of microwave parameters were repeated to validate the findings. A control (no microwave) extract was also run with 100% ethanol and the optimal solvent to powder ratio but where the heat was provided using a water bath instead of the microwave oven. The water bath temperature was chosen to be 76 °C and the incubation was done for 100 s because it was the maximum temperature and incubation time achieved during the MAE using the two sets of microwave parameters. These three extracts were analysed for TSC and antioxidant capacity–measured with two assays, ABTS and DPPH. The energy consumption for these extracts was also estimated according to the Equation (1) as follows:
(1)$$W_{i}=P_{i}\times t_{i}$$
where *W_i_* is the consumed electrical energy for the extraction method (kWh), *P_i_* is the electrical power supplied for the extraction method (kW) and *t_i_* is the electricity consumption time for the extraction method (h).

### 2.3. Analytical Methods

#### 2.3.1. Determination of Total Saponin Content (TSC)

Determination of the total saponin content was conducted using the colorimetric method of Hiai, Oura and Nakajima [26] with slight modifications. The principle of this method is the reaction of sulphuric acid-oxidised saponins with vanillin to produce a distinctive red-purple colour, which is measured at 560 nm using a spectrophotometer.

To 0.25 mL of the appropriately diluted Gac seed ethanol extract samples, 0.25 mL 8% vanillin in ethanol (*w*/*v*) was added followed by 2.5 mL of 72% H_2_SO_4_ (*v*/*v*). The test tube was vortexed, covered, incubated at 60 °C for 15 min and cooled to ambient temperature in an iced-water bucket for 2 min. With a reagent blank as reference, the absorbance was measured at 560 nm using a Carry 50 Bio spectrophotometer (Varian Pty. Ltd., Mulgrave, VIC, Australia).

A standard curve of aescin (100–1000 μg/mL) was constructed to determine the saponin concentrations. The results were expressed as mg aescin equivalents (AE) per gram dry weight of Gac seed kernel powder (mg AE g^−1^).

#### 2.3.2. Determination of Antioxidant Capacity

The antioxidant capacity was tested for the optimal and control extracts using two assays: ABTS and DPPH.

##### ABTS Assay

The ABTS assay [27] was used as described by Tan et al. [28] with slight modifications. Stock solutions of 7.4 mM ABTS and 2.6 mM potassium persulfate were prepared and kept at 4 °C until use. Fresh working solution was prepared for each assay by mixing the 2 stock solutions in equal quantities and incubating them for 15 h in the dark at ambient temperature. Then, 1 mL of the working solution was diluted with ~30 mL of methanol to obtain an absorbance of 1.1 ± 0.02 units at 734 nm. To 0.15 mL of each standard, blank and appropriately diluted extract sample, 2.85 mL of the working solution was added. The tubes were incubated for 2 h in the dark at ambient temperature and the absorption was measured at 734 nm using a Carry 50 Bio spectrophotometer (Varian Pty. Ltd., Mulgrave, VIC, Australia). Trolox was used as the standard and the results were expressed as mg Trolox equivalents per gram dry weight of Gac seed kernel powder (mg TE g^−1^).

##### DPPH Assay

The DPPH assay [29] was used as described by Tan et al. [28]. A stock solution of 0.6 M DPPH in methanol was prepared and kept at −20 °C until use. The working solution was prepared by mixing 10 mL of stock solution with ~45 mL of methanol to obtain an absorbance of 1.1 ± 0.02 units at 515 nm. To 0.15 mL of each standard, blank and appropriately diluted extract sample, 2.85 mL of the working solution was added. The tubes were allowed to stand for 3 h in the dark at ambient temperature and the absorption was measured at 515 nm using a Carry 50 Bio spectrophotometer (Varian Pty. Ltd., Mulgrave, VIC, Australia). Trolox was used as the standard and results were expressed as mg Trolox equivalents per gram of dry weight Gac seed kernel powder (mg TE g^−1^).

### 2.4. Statistical Analyses

Experiments were performed in triplicate and values were expressed as means ± SD and were assessed for statistical significance using the one-way ANOVA and Tukey’s *Post Hoc* Multiple Comparison test using the IBM SPSS Statistics 24 program (IBM Corp., Armonk, NY, USA). Correlation and regression analyses were done using Microsoft Excel 2016. Differences between means, correlations and regressions were considered statistically significant at *p* < 0.05.

## 3. Results

### 3.1. Effect of the Ethanol Concentration on the MAE of Saponins from Full-Fat and Defatted Gac Seed Kernel Powders

The full-fat and defatted Gac seed kernel powders were extracted using MAE with the ethanol concentration ranging from 60% to 100% (in water) in the extraction solvent. Figure 2 shows that at the lower ethanol concentrations, from 60% to 80%, there was no significant difference in the measured TSC for the full-fat powder. The measured TSC was higher with 90% ethanol and the highest (100.3 mg AE g^−1^) with 100% ethanol as the extraction solvent. In contrast, changing the ethanol concentration from 60% to 100% did not increase the measured TSC of the defatted Gac seed kernel powder, which was lower than for the full-fat powder for all the ethanol concentrations. Therefore, the full-fat Gac seed kernel powder and 100% absolute ethanol, as the extraction solvent, gave the best MAE extraction of saponins and they were used in the subsequent experiments.

### 3.2. Effect of the Ethanol to Sample Ratio on the MAE of Saponins from the Full-Fat Gac Seed Kernel Powder

Seven ratios of 100% absolute ethanol to full-fat powder, from 10 to 100 mL g^−1^, were investigated. Figure 3 shows that increasing the ratio from 10 to 30 mL g^−1^ had a significant effect on the measure TSC value after MAE, which increased by 30% from 70.4 to 100.8 mg AE g^−1^. However, increasing the ratio from 30 to 100 mL g^−1^ resulted in less pronounced increases in the measured TSC. Therefore, although the measured TSC was slightly and significantly higher with the ratio of 100 mL g^−1^ compared to 30 mL g^−1^ (Figure 3), the ratio of ethanol to powder of 30 mL g^−1^ was deemed to be the better ratio, from the conservation of solvent perspective, and it was chosen for investigating the microwave parameters. 

### 3.3. Effect of the Microwave Parameters on the MAE of Saponins from the Full-Fat Gac Seed Kernel Powder

Four levels of microwave power (360, 480, 600 and 720 W) were investigated and at every power level, the number of irradiation cycles was also varied (1, 2, 3 and four cycles). Each cycle consisted of 10 s power ON (irradiation) followed by 15 s power OFF (no irradiation). The full-fat powder was used and the ratio of 100% ethanol to powder was 30 mL g^−1^. In general, Figure 4 shows that the measured TSC gradually increased as the power and irradiation time were increased for the MAE but that many of the values were not significantly different from each other. Notably, from 600 W to two cycles upwards (to the right in Figure 4), there was no significant increase in the measured TSC values. However, the two sets of parameters, which only shared the a superscript in Figure 4, 360 W and three cycles and 480 W and four cycles, were selected as possibly optimal for the MAE extraction of saponins from the full-fat Gac seed kernel powder. 

### 3.4. Correlations between the TSC and the MAE Temperature

The temperature of the extracts at the end of each MAE in Figure 4 was recorded using a digital thermometer. Their temperature ranged from 43.4 to 75.6 °C. Correlation analysis revealed that the measured TSC of the extracts was positively correlated with the temperature of the extraction mixture at the end of the MAE (Figure 5).

Table 1 shows that the temperature of the extracts at the end of the MAE was almost all due (92.5%) to the number of irradiation cycles (length of the microwave irradiation time) during the MAE; in contrast, there was no correlation between the temperature and the microwave power. Consisted with this, the measured TSC of the extracts was positively correlated with the number of microwave irradiation cycles but wasn’t correlated with the power used during the MAE. Moreover, there was no interaction between the microwave power and the number of irradiation cycles.

### 3.5. Verification of the Optimal MAE Conditions for the Extraction of Saponins from Full-Fat Gac Seed Kernel Powder

Two possible optimal sets of microwave parameters (360 W and three cycles, 480 W and four cycles) were chosen for the extraction of saponins from the full-fat powder (Figure 4). Notably, the temperature measured at the end of the MAE using the two sets of microwave parameters (360 W and three cycles, 480 W and four cycles) was 72.2 ± 1.2 and 75.6 ± 1.9 °C, respectively, and they were not significantly different from each other.

These two sets of microwave parameters were repeated to validate the findings. A control (no microwave) set of extracts was also run with 100% ethanol and the optimal solvent to powder ratio where the temperature measured at the 480 W and four cycles MAE (76 °C) was provided using a water bath instead of the microwave oven. Also, the time used for the control extraction was chosen to be 100 s in order to match the length of time used for the 480 W and four cycles MAE.

These three sets of extracts were analysed for their saponin content and their ABTS and DPPH antioxidant activities. The results revealed that there was no difference among the three extracts in saponin content and antioxidant capacity (Table 2). However, the ABTS values were low and the DPPH assay did not detect any antioxidant activity for any of these extracts (Table 2).

From the point of view of saving energy, the optimal treatment 1 MAE parameters (Table 2) of 360 W with three irradiation cycles of 10 s power ON and 15 s power OFF per cycle (total of 75 s), were the best microwave conditions for the extraction of saponins from full-fat Gac seed kernel powder (0.003 kWh). The conventional extraction for 100 s in a shaking water bath at the same temperature (76 °C) as at the end of the optimal treatment 2 MAE settings also gave the same results but this also required more energy than the optimal treatment 1 MAE parameters because of the energy needed (0.325 kWh) to bring the temperature of the 5 L water bath from 20 °C up to 76 °C.

## 4. Discussion and Conclusions

The full-fat Gac seed kernel powder was the more suitable material to use as defatting caused a considerable loss of saponins (~75%). The highest TSC of extracts were obtained with 100% absolute ethanol, a 30 mL g^−1^ ratio of ethanol to full-fat Gac seed kernel powder and several sets of MAE conditions. Furthermore, when two sets of the MAE parameters, which gave the highest measured TSC in the extracts, were re-tested, the two extracts had the same TSC. However, from the point of view of saving energy, the optimal 1 MAE parameters of 360 W with three irradiation cycles of 10 s power ON and 15 s power OFF per cycle (total of 75 s), were the best conditions for the extraction of saponins from the full-fat powder.

It was concluded that, to extract Gac seed saponins, it was better to use the full-fat seed kernel powder rather than kernel powder from which the fat had been extracted. The Gac seed saponins appeared to be mainly associated with the fat component of the seeds because they were largely lost during the defatting process with hexane. Undoubtedly, most of the Gac seed saponins (75%) were highly non-polar, which is consistent with a previous finding that saponins were found in the unsaponifiable matter from Gac seed oil [14]. The saponin content of the full-fat Gac seed kernels was also similar to that reported for other oily plant extracts, such as eucalyptus [30] and *Phyllanthus amarus* [31], and significantly higher than the non-oily extract from the flesh of bitter melon [28].

Absolute ethanol was found to be the best concentration of ethanol for extracting the saponins from the full-fat Gac seed kernel powder. This is also consistent with the Gac seed saponins being hydrophobic in nature and consistent with Gac seeds having a high fat content [32]. This result is also consistent with the earlier findings that Gac seed oil has a high content of unsaponifiable matter [33] and that this unsaponifiable material contains triterpenoid saponins [14]. It may be that more non-polar class 3 solvents, such as 1-propanol, isobutyl alcohol and n-butanol, could further improve the extraction of saponins from full-fat Gac seed kernels. However, because of the higher costs of these solvents, ethanol would be the solvent of choice for recovery of the Gac seed saponins on economic grounds.

The ratio of 30 mL ethanol per 1 g of Gac seed powder was the ratio of choice because, at this ratio, the saponin yield was improved significantly compared to the two lower ratios and there was not much improvement at the higher ratios. Although it varies for different plant materials, in the conventional extraction method, the higher the ratio of solvent volume to solid sample the better the extraction of compounds is. However, in the case of MAE, a higher solvent: sample ratio may not necessarily give a better yield due to non-uniform distribution and exposure to microwaves [16].

Varying the MAE parameters did not greatly affect the saponin extraction, mostly likely due to ethanol being a very good solvent [16] for the extraction of the Gac seed saponins and for increasing in temperature even under mild microwave conditions. When the optimal MAE conditions were compared with a control extraction (no microwave), it was found that the same level of saponin extraction was achieved irrespective of the heating source. However, from the point of view of saving energy, the optimal 1 MAE parameters were the best conditions for the extraction of saponins from full-fat Gac seed kernel powder. The energy saving characteristic of MAE has been confirmed in numerous reports [15]. This is due to the heat is generated inside the materials and then comes outwards, whereas in conventional heating the surface is heated first. Thus, the microwave heating is rapid and effective as heat is transferred directly to the material.

The antioxidant capacity of the Gac seed saponin extracts was low. This is possibly due to the more lipophilic nature of the Gac seed saponins, which is consistent with the findings that lipophilic compounds such as carotenoids, which do not show DPPH-radical scavenging activity [34], and tocopherols, which do not show much activity in the ABTS assay [35].

In conclusion, this study demonstrated that the extraction parameters play an important role in the extraction of saponins from Gac seeds. Accordingly, the MAE optimal parameters for the extraction of saponins were a ratio of 30 mL of 100% absolute ethanol per g of full-fat Gac seed kernel powder with the microwave set at 360 W for three irradiation cycles of 10 s power ON and 15 s power OFF per cycle. These parameters are recommended for the extraction of enriched saponins from Gac seeds for potential application in the nutraceutical and pharmaceutical industries.

## Figures and Tables

**Figure 1 medicines-05-00070-f001:**
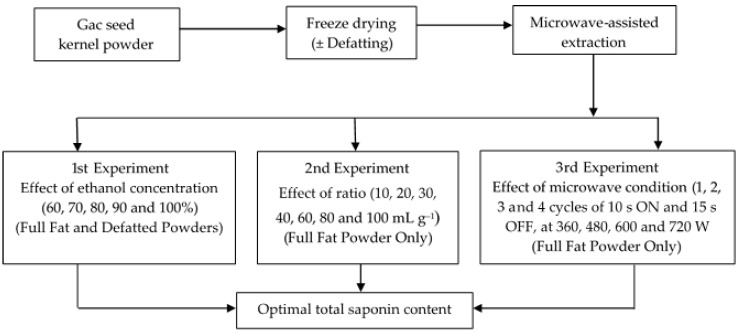
Experimental design for optimisation of saponin yield from Gac seeds.

**Figure 2 medicines-05-00070-f002:**
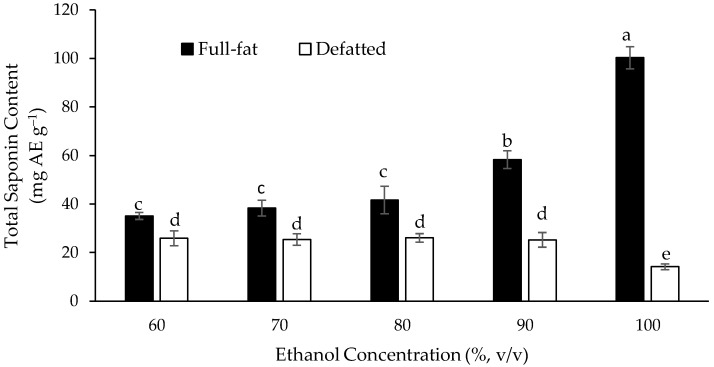
Effect of the ethanol concentration, in the extraction solvent used for microwave-assisted extraction (MAE), on the measured total saponin content (TSC) of the full-fat and defatted Gac seed kernel powders. The values are the means of three replicates for each extraction and columns not sharing the same superscript letter are significantly different at *p* < 0.05.

**Figure 3 medicines-05-00070-f003:**
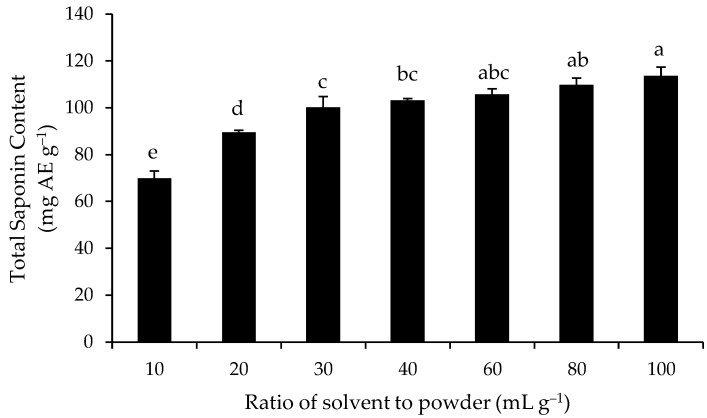
Effect of the ethanol to powder ratio on the TSC of the full-fat Gac seed kernel powder measured using MAE. The values are the means of three replicates for each extraction and columns not sharing the same superscript letter are significantly different at *p* < 0.05.

**Figure 4 medicines-05-00070-f004:**
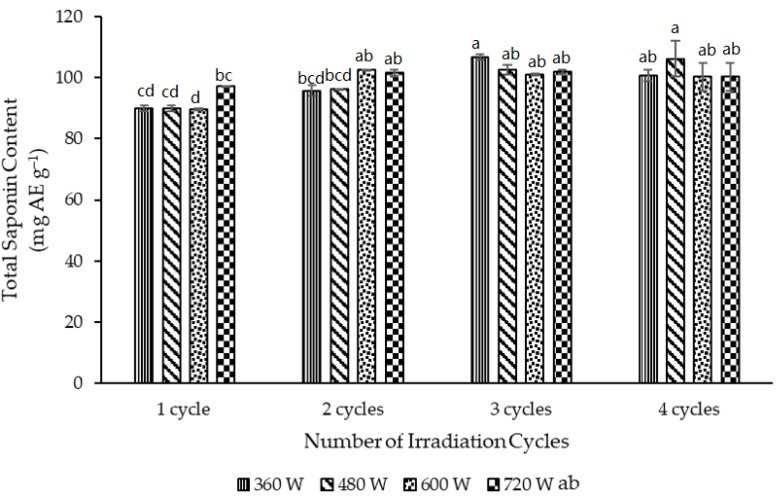
Effect of microwave power and irradiation time (cycles) on the TSC of the full-fat Gac seed kernel powder measured using MAE. The values are the means of three replicates for each extraction and columns not sharing the same superscript letter are significantly different at *p* < 0.05.

**Figure 5 medicines-05-00070-f005:**
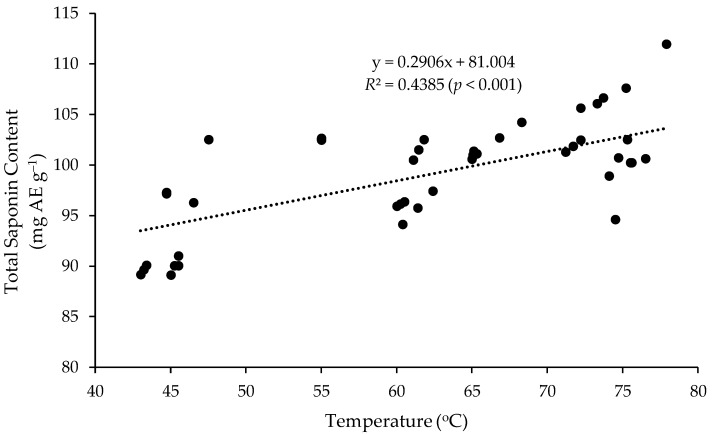
Correlation between the TSC and the temperature of the extract at the end of various MAE treatments. The black dots: TSC at different temperature of the extracts.

**Table 1 medicines-05-00070-t001:** Correlations between the TSC and the MAE parameters.

	*R*^2^ (*p* Value)			
	TSC	Number of Cycle	Power	Cycle and Power
Temperature	0.439 (*p* < 0.001)	0.925 (*p* < 0.001)	0.002 (*p* > 0.5)	0.926 (*p* < 0.001)
Number of cycle	0.362 (*p* < 0.001)	-	-	-
Power	0.000 (*p* > 0.5)	-	-	-
Cycle and Power	0.188 (*p* > 0.1)	-	-	-

**Table 2 medicines-05-00070-t002:** Saponin content, antioxidant activities and energy consumption of the optimal MAE and control extracts.

Extract	TSC (mg AE g^−1^)	ABTS (µmol TE g^−1^)	DPPH (µmol TE g^−1^)	Energy Consumption (kWh)
Optimal treatment 1 ^†^	105.69 ± 2.40 ^a^	1.47 ± 0.12 ^a^	Undetected	0.003
Optimal treatment 2 ^‡^	109.23 ± 2.69 ^a^	1.80 ± 0.31 ^a^	Undetected	0.005
Control (no microwave) ^§^	109.64 ± 4.79 ^a^	1.63 ± 0.10 ^a^	Undetected	0.325

The results are mean values ± standard deviations (*n* = 3) and the values not sharing the same superscript letter in the same column, are significantly different at *p* < 0.05. ^†^ Ethanol + Full-fat powder (30 mL g^−1^); MAE at 360 W, three cycles for 75 s. ^‡^ Ethanol + Full-fat powder (30 mL g^−1^); MAE at 480 W, four cycles for 100 s. ^§^ Ethanol + Full-fat powder (30 mL g^−1^); Shaking water bath at 76 °C for 100 s.

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
