# Peer review of "Optimisation of the Microwave-Assisted Ethanol Extraction of Saponins from Gac (Momordica cochinchinensis Spreng.) Seeds"

_medicines, 2018, doi:10.3390/medicines5030070_

Round 1

Reviewer 1 Report

In this work, authors studied the extraction of saponins from Gac seeds using different parameters (solvent concentration, ratio of solvent to sample and power and time) by Microwave-Assisted Extraction. Moreover, they evaluated the effect of defatting the seed powder on the final extraction of saponins.

MINOR REVISIONS

The manuscript is well structured and written, the conclusions are supported by the analysis of the data presented and therefore the paper can be accepted for publications in “Medicines” although I suggest consider my comments listed below.

-          Abstract:

o   please specify, if correct, that AE is referring to aescin equivalents (line 23).

o   I suggest authors to remove the sentence “MAE or a conventional shaking water bath could be used to obtain asimilar amount of extracted saponins”, as it is explained through the manuscript (lines 25-27).

o   It is not necessary to repeat the conditions stated some lines before, so a change in lines 29-30 is needed, authors could change this sentence by this one: “The optimized MAE conditions are recommended for the extraction of enriched saponins from Gac seeds for potential application in the nutraceutical and pharmaceutical industries”.

-          Introduction section:

o   misspelled word in line 35 (introduction instead of “introdution”)

o   Lines 51-52 repeat the same as it was said in lines 45-46. Decide what sentence use in order to avoid repetition.

o   Objective: I suggest to use this sentence in lines 75-76: “Therefore, in this study, the extraction of saponins from powdered Gac seed kernels was optimised using MAE with ethanol as the extraction solvent.”

-          Materials and methods:

o   Lines 97-99 (including figure 1) should be moved to section 2.2., as it includes information regarding this specific section.

o   Lines 131-132 should be deleted. This sentence is relevant in “Results” section, not in this part of the manuscript.

o   Line 148. I suggest to add “(no microwave)” to this sentence: “A control (no microwave) extract…” to help the reader to understand quickly the importance of this specifici control assay.

o   Line 164: include here information about spectrophotometer used (“using a Carry 50 Bio spectrophotometer (Varian Pty. Ltd., Mulgrave, VIC, 171 Australia)”). In the same line, use only this sentence “A calibration curve was constructed (1,000 - 100 μg/ml) to determine the saponin concentrations”. Do not include the regression line.

o   Delete lines 167-178, as this is information not relevant. It is well specified that results are expressed as mg AE/g in lines 179-180, so authors don’t need lines 167-178.

o   Line 207: Add city and country for IBM SPSS Statistics program.

-          Results:

o   Lines 264-270: Only write “Table 1” once, as all the paragraph is referring to this table. Delete Table 1 that appear in brackets.

o   Line 274: Delete “From the findings in figure 4”, and include “(Figure 4)” at the end of the sentence.

o   Line 279: Include “A control (no microwave)” to better understanding.

o   In table 2, authors could include also this to the assays regarding to the control. For example, “Control (no microwave)”. I suggest them to use “Optimal treatment 1” and  “Optimal treatment 2”, and refer to them by using the same term through lines 294-300.

-          Discussion and conclusion:

o   Do not use the same paragraph in lines 302-306 than that used in lines 75-79.

o   Line 307: Delete “First, it was found that” and “(Figure 2), (Figure 3 and 4), (Table 2), etc”, as they should be used only in Results section. Directly say that “The full-fat Gac seed kernel…”

o   Lines 339-346: The authors refer to a control extraction. They should highlight and better discuss the importance of using microwave, as these waves imply a less energy procedure to obtain the same amount of extracted saponins. The lowest energy usage is a factor justifying the use of microwave.

Author Response

Please find attached the response to the Reviewer 1 comments.

Reviewer 2 Report

It would be interesting to know the composition of the saponin extract.

The article describes the search for extraction conditions for saponins from  Gac  (Momordica  cochinchinensis  Spreng)  seed kernels. They use the method of Hiai, Oura and Nakajima for determination of saponine contents, I would rather prefer to have a structural determination of the saponins isolated or at least a NMR of the crude extracts.

Author Response

Please find attached the response to the Reviewer 2 comments.

Reviewer 3 Report

The manuscript “Optimisation of the microwave-assisted ethanol extraction of saponins from Gac (Momordica cochinchinensis Spreng) seeds” was submitted to Medicines for publication. The study describes the validation of an extraction procedure and the influence of different parameters on the yield of triterpenoids.

Broad comments:

The whole validation procedure is very well described and the fact that defatting of the seeds leads to a decrease in the yield of saponins sounds reasonable, as lipophilic constituents are known to enhance the saponins’ solubility.

However, the authors claim that “absolute ethanol was found to be the best solvent for extracting the saponins from the full-fat Gac seed kernel powder” (line 324-325), which is only partly true, as only ethanol (in different concentrations) was used for extraction. Although it is true that ethanol is to be preferred over methanol in terms of toxicity, a number of class 3 solvents is suggested by the FDA. Thus, at least two other organic solvents should be tested for their extraction potential, such as e.g. acetone or isopropanol, whereof the latter is a common solvent for triterpenoid extraction.

Furthermore, the authors state that “microwave heating is able to disrupt the plant cell structure” (line 57), which is also the fact for ultra-sonication, another very common and economic extraction technique. Therefore, comparison of microwave extraction to ultra-sonication would be of greater value for this study and of much more interest to the readership.

The last finding is that the authors use a hydrolytic solvent in combination with high temperatures, what raises the question for the compounds’ stability. Even though irradiation times are kept to a minimum, temperature increases to 76°C should not be ignored and information on the eventual degradation would be important.

Additional comments:

The ABTS and DPPH assays are out of scope for this topic and are rather lowering the soundness of an actual interesting study.

Author Response

Please find attached the response to the Reviewer 3 comments. 

Round 2

Reviewer 3 Report

Please give the name/model number of the microwave used.

Author Response

Dear Reviewer,

The authors thank you for your comment to improve the manuscript. 

Your comment "Please give the name/model number of the microwave used" has been addressed by adding the model of the microwave to the manuscript (Line 121).

Kind regards,

Anh Le